# Analysis of Food Safety Management Systems in the Beef Meat Processing and Distribution Chain in Uganda

**DOI:** 10.3390/foods10102244

**Published:** 2021-09-22

**Authors:** Siya Balaam Jeffer, Issmat I. Kassem, Samer A. Kharroubi, Gumataw Kifle Abebe

**Affiliations:** 1National Food Safety Foundation (NFSF), The Affiliated Institution of the Food Safety Associates Limited, Kampala P.O. Box 2244, Uganda; Balaamjeffer@gmail.com; 2Center for Food Safety, Department of Food Science and Technology, University of Georgia, 1109 Experiment Street, Griffin, GA 30223-1797, USA; issmat.kassem@uga.edu; 3Department of Nutrition and Food Sciences, Faculty of Agricultural and Food Sciences, American University of Beirut, Beirut 1107-2020, Lebanon; sk157@aub.edu.lb; 4Department of Business and Social Science, Faculty of Agriculture, Dalhousie University, Truro, NS B2N 5E3, Canada

**Keywords:** food safety management, meat hygiene practices, beef supply chain, Uganda

## Abstract

Meat production is an essential component in food security and the economy in Uganda. However, food safety concerns pose a challenge to public health in Uganda and impede access to regional and global markets. Here, food safety management (FSM) practices in the Ugandan beef supply chain were evaluated. A cross-sectional survey was conducted in major slaughterhouses (*n* = 3), butcher shops (*n* = 184), and supermarkets (*n* = 25) in Uganda’s capital, Kampala. The three slaughterhouses had low scores in core control and assurance activities of FSM. Packaging interventions were weak in all the slaughterhouses, while only one slaughterhouse had a functional cooling facility. Supermarkets implemented better hygienic and preventative practices in comparison to butcher shops. However, both sourced from slaughterhouses that had low-to-poor hygiene practices, which weakened the efforts implemented in the supermarkets. Furthermore, most butcher shops did not offer training to meat handlers on HACCP (Hazard Analysis and Critical Control Point)-based practices. The low food safety performance in the supply chain was primarily attributed to poor sanitation, hygiene, and handling practices. Therefore, HACCP-based training and robust preventive, intervention, and monitoring systems are needed in the Ugandan beef supply chain to benefit public health and increase competitiveness.

## 1. Introduction

Due to the increase in population, urbanization, and income, demand for meat has surged globally. This trend has also been observed in Uganda, where meat production is considered an essential component of the economy and food security [1]. According to the Ugandan Ministry of Agriculture, Animal, and Fisheries and the Uganda National Bureau of Statistic survey (2018) [2], Uganda is estimated to have over 14 million cattle, 16 million goats, 4 million sheep, 47 million chickens, and 4 million pigs. About 4.5 million households (70.8%) farm at least one kind of livestock or poultry [3,4]. The indigenous breeds dominate the Ugandan cattle production, which is primarily considered to be an extensive system. Furthermore, the meat sector contributes about 9% of Gross Domestic Product (GDP) and 17% of the agricultural GDP [5]. In Uganda, meat consumption is the highest in the capital and the largest city of Uganda, Kampala, where demand for beef is estimated to be 15,500 tons annually [5]. Taken together, these observations highlight the importance of the meat supply chain for the economy and as a source of protein for the population.

In the Ugandan beef supply chain, most farmers sell their animals at the local market or farm gate to a middleman, who then transports and sells the animals to slaughterhouses or the carcasses to retailers. At slaughterhouses, animals are usually kept for 2 to 10 days, depending on beef demand and inspection, to satisfy required animal health standards. After slaughtering, a veterinarian will stamp the carcass to indicate that the meat is deemed safe. Apart from slaughterhouses, there are several locations in Uganda where animals are slaughtered, including farm gates, village markets, and town slaughter slabs. Most food manufacturers buy meat at the slaughterhouses and process the meat into various products such as prime cuts, minced meat, and sausages. Meat and meat products are exported in small quantities to South Sudan, the Democratic Republic of Congo, and, more recently, Somalia. Additionally, a small number of live animals are exported to South Sudan [1,6]. Currently, the beef supply chain is primarily limited to domestic consumption due to noncompliance with the food safety requirements of international markets [6]. Uganda has continuously failed to supply meat to more lucrative markets in Europe, the Middle East, and China because of quality and safety issues [1]. Therefore, while meat production is essential, it also appears to be challenged by quality and safety issues that impede taking full advantage of this vital sector in Uganda. Failure to export beef is predicted to result in potential economic losses, while poor meat hygiene and preventative practices adversely affect public health and increase the cycle of disease and poverty locally.

Previous studies have reported a high prevalence of zoonotic livestock-associated diseases in Uganda [7,8,9]. Notably, the consumption of contaminated beef was associated with an outbreak of gastrointestinal anthrax in the Isingiro District, Uganda, in 2017. Epidemiological analysis showed contaminated meat bought from butcheries being the leading cause of the outbreak. Another study also detected an unacceptable level of microbial contamination in meat samples collected from abattoirs and butcheries in Kampala [10]. This finding was corroborated by a similar study that concluded poor hygienic standards and handling practices of beef in slaughterhouses and butcheries [11]. These studies called for more stringent food safety practices and monitoring programs in Uganda. Although less than a handful of studies have highlighted concerns on meat safety, investigations of food safety management practices at the level of the beef supply chain remain scant in Uganda [12,13,14,15,16]. This is very important because a supply chain perspective can shed light on the food safety culture and hygienic practices of various actors in the meat supply chain [17]. Against such a backdrop, this study is conducted to examine food safety management practices at the level of the supply chain that included slaughterhouses, butcher shops, and supermarkets in Kampala, which is the principal local government administrative unit and has five divisions—Rubaga, Kawempe, Nakawa, Makindye, and Central. Each division has at least one primary market that is under the jurisdiction of the Kampala City Council Authority (KCCA), which is responsible for ensuring that the slaughtered animals are safe for human consumption, while the Ministry of Health (MOH) and the Uganda National Bureau of Standards (UNBS) officials are authorized for controlling the hygiene and sanitation programs at the slaughterhouses. Figure 1 provides an overview of the Ugandan meat supply chain.

In this study, we focused on food safety practices—including sanitation, hygiene, and handling—of the key supply chain actors that provide beef to consumers in Kampala. For this purpose, we designed cross-sectional questionnaires to survey food safety and hygienic practices in slaughterhouses and retail outlets (butcher shops and supermarkets) because these locations are highly susceptible to contaminations that can result in severe foodborne disease outbreaks [18]. To our knowledge, this is the first study that assesses food safety practices in the beef supply chain in Uganda. The findings of this study have important implications and can enhance public health and the competitiveness of the Ugandan beef supply chain in local and global markets.

## 2. Materials and Methods

### 2.1. Sampling Approach

There are five major slaughterhouses in Kampala. All of them were invited to participate in the study, but only three slaughterhouses participated. The exact number of butcher shops in Kampala could not be found from the Kampala City Council Authority (KCCA). Nonetheless, we estimated a total of 525 butcher shops based on the number of primary markets across the five divisions of Kampala. To ensure the representation of butcher shops from the five divisions, we randomly selected 15 butcher shops from each major market including Nakasero, Usafi, Makindye, Kalerewe, Wandegeya, Nakawa, Bugolobi, Kireka, Bweyogerere, Kisekka, Katwe, Mpelerwe, Kitintale, Luzira, and Natete. Of the 225 invited butcher shops, 196 butcher shops (87.11%) were able to participate in the study. Likewise, we were able to identify and invite 46 supermarkets that regularly sell beef to consumers in and around Kampala, including Quality supermarket, Tuskys, Capital Shoppers, Shoprite, Uchumi, Mega standards, and Kenjory. A total of 25 supermarkets (54.35%) participated in the study.

Two structured questionnaires (full questionnaires are available in the Appendix A), approved by the Institutional Review Board (IRB) at the American University of Beirut (AUB), were used to collect data. The first questionnaire was related to slaughterhouses and included several indicators that measured food safety control activities (such as preventive measures design, intervention system design, monitoring system design, and actual operation of control strategies), assurance activities (validation, verification, documentation, and record-keeping systems), and performance. The indicators were adopted from the food safety management systems diagnostic instrument (FSMS-DI) developed by Jacxsens and Luning [19,20], which systematically analyzes the degree to which core control and assurance activities are implemented. FSMS-DI has been used previously in assessing FSMS in meat, poultry, dairy, fish, and lamb supply chains [21,22,23,24,25]. The second questionnaire was used to establish the sanitation and hygiene practices deployed by the butcher shops and supermarkets. The questionnaire was based on ‘Butcher Safe’ [26], which is a guideline developed by the Scottish Food Enforcement Liaison Committee (SFELC), HACCP (Hazard Analysis and Critical Control Point) Working Group, to help butchers comply with the HACCP requirements. This guideline has been used previously to evaluate food safety management systems in butcheries in Belgium, Scotland, and the UK [27]. The questions included methods for cleaning and sanitizing, frequency of premises cleaning, pest control, waste management, maintenance of premises and equipment, personal hygiene, staff training, and the availability of standards.

### 2.2. Data Collection and Analysis

Face-to-face interviews and on-site visits were carried out with the quality control officers or managers of slaughterhouses, the owners of the butcher shops, or persons responsible for the meat section of the supermarkets. The interviews were administered in person by one of the coauthors, a native speaker of the local language. A written consent form was handed to the participant to inform them on the research topic and the approach used, including their right to withdraw from the study. If participants did not understand English, the interviewer communicated with those participants in the local language (Luganda). No personal identifiers were collected in the study. Data collection was carried out between 18 December 2019 and 4 February 2020.

The core control and assurance activities and food safety performance of slaughterhouses were qualitatively measured and transformed into assigned ratings, as described in Luning et al. [28]. The scores represent qualitative descriptions ranging from zero (not applied or nonexistent) to three (advanced food safety activities/performance). Similarly, the responses of butcher shops and supermarkets were entered into Microsoft Excel 2016 and imported to IBM SPSS (Version 23.0) for analysis. Statistical analysis was performed to obtain the mean scores of the core control and core assurance activities and the food safety performance of slaughterhouses.

## 3. Results and Discussion

### 3.1. Description of Slaughterhouses and Beef Retail Outlets in Kampala

We documented that each slaughterhouse slaughtered 200 to 400 animals, on average, daily. The slaughterhouses were located close to heavy traffic and residential houses. Furthermore, the fencings were inadequate to prevent vermin and unauthorized people from entering the slaughterhouses. The butcher shops were mainly concentrated in division markets; only a few were located on the roadsides. All the supermarkets and butcher shops in the central markets had permanent structures, while most roadside butcher shops had semipermanent structures. Most supermarkets (75%) reported having sourced meat from Uganda Meat Parkers and/or City Abattoirs. However, most butcher shops did not have a specific slaughterhouse for sourcing; only 40% reported using one slaughterhouse. In the study context, male workers dominated the meat retail outlets.

### 3.2. Food Safety Management System (FSMS) Performance of Slaughterhouses

#### 3.2.1. Core Control Activities

As shown in Table 1, the average score for the core control activities among the slaughterhouses was one. This suggested that control activities were basic, often characterized by minimal criteria used for FSMS evaluation in addition to various food safety problems.

The preventative design measures of slaughterhouses had a mean value of 2, indicating that the slaughterhouses applied expert knowledge, governmental guidelines, best practices, or standardized methods to prevent problems that may occur occasionally. The control activities for sanitation programs, personal hygiene requirements, and animal control scored better than other metrics, possibly because the slaughterhouses have had regular visits from government veterinarians and inspectors from KCCA, UNBS, and MOH. However, only one slaughterhouse had a functional cooling facility, but it was not consistently used due to fluctuating electricity and high energy costs. Furthermore, the environmental temperature of the cooling facility was not automated, and the facility was old (built in the 1970s). The lairages and kraals were mainly open areas with insufficient shelter to prevent the animals from the harsh climate, leading to dehydration and exposure to various pathogens and contaminants via vectors and the environment. Furthermore, of the three slaughterhouses, only one had a sheltered lairage and kraal, and it needed renovation. The kraals of the slaughterhouses had not been partitioned to separate incoming animals from stock or to isolate sick animals. Additionally, there were inadequate procedures in place to prevent the cross-contamination of carcasses. The slaughtering and skinning of animals were mainly performed using knives and machetes rather than electric cutters.

Due to the inadequate availability of running water and electricity in the slaughterhouses, physical intervention practices such as carcass trimming and washing, hide and offal washing, and equipment sterilization were rare. Instead, they applied a hot water rinse to sterilize equipment; however, two of the slaughterhouses did not have thermometers to monitor the temperature of the water used for sterilization during the site visit. Commonly recommended sterilization methods such as acetic acid and lactic acid rinses [29] have not been adopted, possibly due to the associated costs. Packaging intervention equipment was nonexistent in all three slaughterhouses. This might be because the slaughterhouses would often sell the carcasses immediately to retail outlets. Sometimes, the slaughterhouses would place the carcasses in wooden boxes or wrapped with polyethylene for transportation; these packages are not designed to reduce or inactivate potential pathogens and do not maintain a proper cooling temperature [30]. The slaughterhouses’ monitoring systems were generally poor (score 1) and lacked quality control laboratories for microbial and chemical analysis. The veterinarian mainly focused on observing the internal carcass organs for gross infections.

The design monitoring system of the slaughterhouses was basic, with a mean score of 1 (Table 1). Two of the slaughterhouses reported having Critical Control Points (CCPs) based on general hygiene codes. However, no labels indicated the CCPs points along the processing line, and they were not readily recognizable apart from a few activities such as postmortem inspections. Measuring equipment varied between the three slaughterhouses. While one slaughterhouse had an in-line automated measurement with a visual information history, it was not fully functional and had been in place since the 1970s. In addition, the slaughterhouses had not fully adopted a calibration program for measuring and analytical equipment. However, one slaughterhouse had an advanced program (level 3) outsourced to UNBS, which helped calibrate the weighing scales.

There were minimal corrective actions in the slaughterhouses (Table 1). Two slaughterhouses had some procedures but they were difficult to understand, paper-based (were not updated), and/or had not been digitized. Similarly, compliance with set procedures was basic or not followed thoroughly. Slaughtering, bleeding, skinning, and evisceration were performed in the same area. A concern was that drainage from the slaughtering area passed through the kraals, increasing the chances of animal cross-infection [31,32]. The heads, legs, and skins from the carcasses would generally be left on the floors of the slaughterhouses. One slaughterhouse had planned corrective procedures, and the operators were aware of the existence and content of the procedures and followed them correctly. Furthermore, safety tasks were adopted and employees exercised self-control in compliance with procedures.

#### 3.2.2. Core Assurance Activities

Core assurance activities are related to the definition of system requirements, validation, verification, documentation, and record-keeping. These activities offer evidence of meeting food safety requirements [28,30] and provide confidence to various stakeholders [33]. As shown in Table 2, the slaughterhouses did not fully implement the core assurance activities (average score 1). This could be because the focus of food safety regulations and guidelines in Uganda, such as the Public Health Act 1964, the 2003 National Meat Development Policy, and the Meat and Milk Hygiene Regulation, has been primarily on control activities [34]. Further, core assurance activities require resources to implement.

The slaughterhouses had low-to-average scores regarding the sophistication of validation of preventive measures, intervention, and monitoring systems (Table 2). Additionally, the three slaughterhouses had zero-to-low scores regarding the verification of performance related to the people, equipment, and methods (Table 2). The slaughterhouses did not have quality assurance departments to implement preventive measures. Documentation and record-keeping in the three slaughterhouses were also deficient. Since employees have limited access to the latest data and information, they would depend more on experience and less on science-based decisions [34]. This would prevent employees from fully engaging in the management of food safety.

#### 3.2.3. Food Safety Performance of Slaughterhouses

A high level of core control and assurance activities is associated with better food safety performance [19]. However, given that the slaughterhouses did not score highly in core controls and assurance activities, it was expected that the slaughterhouses had poor food safety performance (Table 3). This corroborated the findings of Bogere et al. [10], who found a high level of microbial contamination in slaughterhouses in Kampala, which raises public health concerns.

### 3.3. Assessment of Hygiene Practices in Butcher Shops and Supermarkets in Kampala

Most of the sampled butcher shops implemented basic hygienic practices. Table 4 provides a comprehensive assessment of hygienic practices (pest control, waste management, personal hygiene, environmental hygiene, carcass transportation, storage, and staff training) and government oversight in the butcher shops and supermarkets in Kampala.

Only 15% and 39% of the butcher shops had rodent traps and electronic fly devices, respectively, and about half of them kept the meat in pest-proof containers (Table 4). Nearly three-quarters of the butcher shops controlled pests by keeping the floors, walls, roofs, doors, and windows in good working conditions and leaving no gaps or spaces. This might be because the UNBS Meat and Milk Hygiene Regulation requires butcheries to implement these practices. In comparison, the supermarkets had a superior performance on all aspects of pest control measures. This is probably because supermarkets target the middle to a higher-income population who are likely more conscious of food safety hazards. Therefore, butcher shops pose a higher risk of microbial contamination of meat from pests, especially houseflies [14].

Personal hygiene is paramount when handling meat because there is a high possibility that meat handlers could be vehicles for contaminating meat with pathogenic microorganisms that can cause foodborne diseases [35]. Both the butcher shops and supermarkets generally had good personal hygiene practices (Table 4). Most of the butcher shops and all the supermarkets reported excluding workers in cases of illness. However, none of the meat outlets used protective clothing (e.g., gloves). Personal protective equipment (e.g., clothes, gloves, and gumboots) continues to be a major problem among meat handlers in Africa [12,13,16,35,36]. In this study, all meat handlers used the same bare hands with which they held meat to receive money, increasing the risk of meat contamination. The latter was similar to the findings of Muinde et al. [37], who reported these practices among street food vendors in Kenya. A study by Todd et al. [38] attributed these poor hand hygiene practices to lack of time, inadequate facility and supplies, lack of accountability, and commitment. Notably, meat handlers in the butcher shops lacked handwashing facilities such as running tap water, washing basins, and soap.

All supermarkets used trucks—refrigerated (75%) and unrefrigerated (25%)—for transportation, while only a third of butcher shops used trucks—refrigerated (10%) and unrefrigerated (25%). Furthermore, about 7% of the butcher shops used human labor for transporting meat, while others used motorbikes. Therefore, butcher shops appeared to pose a high risk of meat contamination. This supported Bogere et al. [10], who reported a high prevalence of bacterial loads at the butcher shops in Kampala. A preference for motorbikes by the retail outlets may be due to the increased traffic in the city and the associated costs and benefits. The wide use of motorbikes for transportation of meat carcasses would likely increase food safety risks. Although supermarkets tend to use trucks more frequently, those vehicles are also used for other food and nonfood items, increasing the risk of microbial cross-contamination. There is also inadequate cleaning of the trucks, with bloodstains persisting from previously transported meat carcasses.

Retail outlets in Kampala reported having some measures to counter cross-contamination from the environment. For example, they stored meat on hangers to avoid contamination and ensure that the floors and walls were impervious and thoroughly cleaned and disinfected. They also restricted entrance by nonmeat handlers into areas where meat is stored. However, not all butcher shops used hangers for meat storage; instead, many placed the beef on counters and put wooden stamps on it, leading to a higher risk of meat contamination. Additionally, some butcher shops with hangers did not have protective glass to reduce contamination from houseflies/pests and dust. Furthermore, the butcher shops mixed meat with offals on tables. About two-thirds of the butcher shops and at least 90% of the supermarkets stored beef in glass compartments and refrigerators. Overall, the supermarkets had better storage practices compared with the butcher shops. This might be attributed to the high cost of energy and refrigerators. As a result, most of the butcher shops either would have to stock the beef only for a day or share storage facilities with other butcher shops nearby. However, we found about 34.2% of the butcher shops kept the beef for more than a day, which increases the chance of contamination. Further, most of the butcher shops located along the dusty streets of Kampala hang the meat in an open space, and thus, have a higher risk of environmental contamination from dust and flies.

Most of the butcher shops provided training for new hires; however, they did not have retraining and/or HACCP-based training programs in place. The lack of HACCP-based training was reflected by the poor food safety management practices of the butcher shops. They did not have optimal record keeping, sanitation, and hygiene practices. Butcher shops mainly focused on specific meat handling practices related to cutting meat parts, customer service, and payment transactions. The mentors also did not have HACCP-based training, instead relying on their experience. In comparison, the supermarkets performed better in retraining and HACCP-based training practices, likely due to the hiring of better-qualified personnel (with some experience).

Government oversight of the butcher shops was very poor. Only 8% of the butcher shops reported on-site inspection by government authorities; moreover, only 2% of the butcher shops were aware of the specific requirements of the government to run or open meat retail outlets. Both the butcher shops and supermarkets reported having received no training support from the government. Apparently, on-site inspections were focused on supermarkets; about 75% of them reported their meat handling practices being inspected by the government, while 20% only claimed that they were aware of specific government requirements to run or open a meat retail outlet. The poor government oversight or lack of support could be due to the lack of coordination among the different government ministries in charge of the meat sector, which include MAAIF; UNBS; the Ministry of local government; the Ministry of Trade, Industry, and Cooperatives (MTIC); and MOH (MAAIF, 2012). These institutions have fragmented and overlapping efforts and jurisdictions; thus, they are unable to provide the necessary support and enforce proper hygiene practices in meat retail outlets in Uganda.

## 4. Conclusions

Meat safety in Uganda continues to be a public health concern and a major development challenge. Although some food-safety-related measures have been implemented to address this issue, meat-associated disease outbreaks and the rejection of meat in global markets continue to occur. This study aimed to explore beef hygiene practices from a supply chain perspective. Therefore, we assessed the food safety measures in slaughterhouses, butcher shops, and supermarkets in Kampala. The study revealed that the slaughterhouses lacked the minimum meat hygiene practices to address basic public health concerns. Furthermore, butcher shops in Kampala did not adhere to the required sanitation and hygiene standards, while supermarkets were better in all aspects of meat hygiene practices. However, it should be noted that supermarkets source from slaughterhouses or butcher shops that have low to poor hygiene practices, which weakens the efforts implemented in the supermarkets. Therefore, unless there is a coordinated effort along the beef supply chain, better meat handling practices in one segment of the chain (e.g., supermarkets) do not guarantee food safety, increasing public health risks.

While this study provides valuable insights into food safety management practices within the beef supply chain in Uganda, there are some limitations that can be addressed in future studies. For example, future studies should expand and corroborate the qualitative assessments of slaughterhouses and meat retail outlets with laboratory analyses. Finally, the study focused on Kampala, Uganda’s capital and trade hub. Future studies should expand to other cities in Uganda.

Several steps can be adopted by stakeholders to enhance food safety in the beef meat chain in Uganda: (1) the slaughterhouses need guidance to adopt science-based prevention, intervention, and monitoring systems—perhaps via bolstering governmental infrastructure in this area; (2) the adoption of effective FSMS and third-party certifications might be critical to compete in regional and international markets; (3) there is a need to promote a meat hygiene culture in slaughterhouses and butcher shops particularly, and across the beef supply chain in general, via outreach and accessible educational material. In that regard, government oversight needs to focus on providing training supports from the supply chain perspective, not only via inspections targeting specific slaughterhouses and meat retail outlets.

To the best of our knowledge, there are limited studies that have applied the Food Safety Management Systems diagnostic tool (FSMSDI) to understand the core control and core assurance activities of slaughterhouses in the context of Sub-Saharan Africa. Subsequently, this study expands our understanding of the meat hygiene practices in Uganda from a supply chain perspective and the complexities of food safety management practices.

## Figures and Tables

**Figure 1 foods-10-02244-f001:**
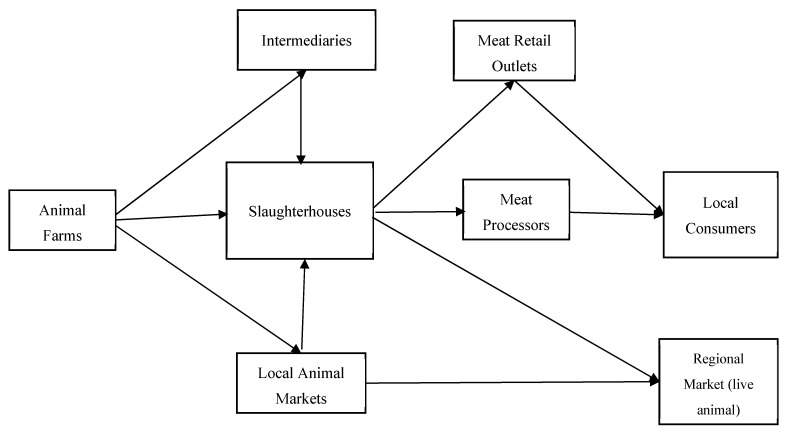
Overview of the Ugandan beef supply chain (source: own description).

**Table 1 foods-10-02244-t001:** Analysis of the core control FSMS activities at the three largest slaughterhouses in Kampala.

	Frequency of Individual Scores of all Three Slaughterhouses				Mean Scores (Assigned)	
Indicators of FSMS activities	**0**	**1**	**2**	**3**		
Core safety control activities						**1**
Design preventive measures					2	
Sophistication of hygienic design of equipment and facilities	0	2	1	0	1.3 (1)	
Adequacy of cooling facilities	2	0	0	1	1 (1)	
Specificity of sanitation program	0	0	3	0	2 (2)	
Extent of personal hygiene requirements	0	0	2	1	2.3 (2–3)	
Adequacy of raw material control	0	0	2	1	2.3 (2–3)	
Specificity of product specific preventive measures	0	0	3	0	2 (2)	
*Design intervention processes*					1	
Adequacy of physical intervention equipment	1	0	2	0	1.3 (1–2)	
Adequacy of packaging intervention equipment	3	0	0	0	0 (0)	
Specificity of maintenance and calibration for (intervention) equipment	0	2	1	0	1.3 (1–2)	
Specificity of intervention methods (chemical and biological)	0	2	1	0	1.6 (1–2)	
*Design monitoring system*					1	
Appropriateness of CCP analysis	1	0	2	0	1.3 (1–2)	
Appropriateness of standards and tolerances design	1	1	1	0	1 (1)	
Adequacy of analytical methods to assess pathogens	2	1	0	0	0.3 (1)	
Adequacy of measuring equipment to monitor the critical process and product conditions	0	1	1	1	2 (2)	
Specificity of calibration program for measuring and analytical equipment	0	2	0	1	1.6 (1–2)	
Specificity of sampling design (microbial assessment) and measuring plan	2	1	0	0	0.3 (1)	
*The extent of corrective actions*					0	
Operation control strategies	0	2	1	0	1.3 (1–2)	
Actual availability of procedures	1	1	1	0	1 (1)	
Actual compliance to procedures	0	1	2	0	1.6 (1–2)	
Actual hygienic performance of equipment and facilities	0	0	3	0	2 (2)	
Actual cooling capacity	2	0	1	0	0.6 (1)	
Actual process capability of physical intervention equipment	1	1	1	0	1 (1)	
Actual process capability of packaging intervention equipment	3	0	0	0	0 (0)	
Actual performance of measuring equipment	0	2	0	1	1.6 (1–2)	
Actual performance of analytical equipment	3	0	0	0	0 (0)	

Note: Scores in bold are the overall scores for core control activities. The scores in brackets are the mean scores for each activity. If an average score for an activity is between 0 and 0.2, the allocated score is 0; if between 0.3 and 1.2, score is 1; if between 1.3 and 1.7, score is 1–2; if between 1.8 and 2.2, score is 2; if between 2.3 and 2.7, score is 2–3; if between 2.8 and 3.0, score is 3. For the indicators of FSMS activities, 0 indicates a low level (absence or not applied), 1—basic level, 2—average level, 3—advanced level.

**Table 2 foods-10-02244-t002:** Analysis of the core assurance FSMS activities at the three largest slaughterhouses in Kampala.

	Frequency of Individual Scores of All Three Slaughterhouses				Mean Scores (Assigned)	
Indicators of FSMS activities	**0**	**1**	**2**	**3**		
Core assurance activities						**1**
Defining system requirements					1	
Sophistication of translation of external requirements into FSMS	1	1	1	0	1 (1)	
Degree of systematic use of feedback information to advance FSMS	1	1	1	0	1 (1)	
*Validation*					1–2	
Sophistication of validation of preventive measure	0	1	2	0	1.6 (1–2)	
Sophistication of validation of intervention systems	0	1	2	0	1.6 (1–2)	
Sophistication of validation of monitoring system	0	1	2	0	1.6 (1–2)	
*Verification*					0	
Extent of verification of people-related performance	2	0	0	1	1 (0)	
Extent of verification of equipment and methods-related performance	2	0	1	0	0.6 (1)	
*Documentation and record-keeping*					1–2	
Appropriateness of documentation system	1	0	2	0	1.3 (1–2)	
Appropriateness of record-keeping system	0	2	1	0	1.3 (1–2)	

Note: Scores in bold are the overall scores for core assurance activities. The scores in brackets are the mean scores for each activity. If the average score for an activity is between 0 and 0.2, then the allocated score is 0; if between 0.3 and 1.2, score is 1; if between 1.3 and 1.7, score is 1–2; if between 1.8 and 2.2, score is 2; if between 2.3 and 2.7, score is 2–3; and if between 2.8 and 3.0, the score is 3. For the indicators of FSMS activities, 0—low level (absent or not applied), 1—basic level, 2—average level, 3—advanced level.

**Table 3 foods-10-02244-t003:** The food safety performance of the slaughterhouses.

	Frequency of Individual Scores of All Three Slaughterhouses				Mean Scores (Assigned)	
Indicators of performance output	**0**	**1**	**2**	**3**		
*Food safety performance*						**1**
Food Safety Management System evaluation	0	3	0	0	1 (1)	
Seriousness of remarks	2	0	1	0	0.6 (0–1)	
Microbiological food safety complaints	2	0	1	0	0.6 (0–1)	
Hygiene-related complaints	0	2	1	0	1 (1)	

Note: Score in bold is the overall food safety performance score of all the slaughterhouses in Uganda; 0—no indication of performance, 1—poor performance, 2—moderate performance, 3—excellent performance.

**Table 4 foods-10-02244-t004:** Assessment of hygiene practices carried by the butcher shops and supermarkets.

Variable	Butcher Shops (*n* = 184) Percentage (% Yes)	Supermarkets (*n* = 20) Percentage (% Yes)
*Pest control*		
Presence of rodent traps (e.g., rat trap, housefly traps, fly screens)	39.1	90
Keeping the floors, walls, roof, doors, and window openings in a good state of repair with no gaps or spaces	72.3	95
Keeping the meat in pest-proof containers	51.6	90
Electronic fly device	14.7	75
*Waste management*		
Waste (inedible parts) is placed in containers with suitably fitting lids and removed frequently from meat handling areas where it is produced	71.2	65
Use of waste containers but without lids	25.5	70
Waste containers regularly cleaned and disinfected	87	85
Presence of a waste control plan	26.1	75
*Personal Hygiene*		
Hand washing	94.6	95
Use of protective clothing such as gloves	0	0
Exclusion from work in case of illness	84.2	100
Reporting of illness	47.8	75
*Environmental hygiene*		
Designate the area where meat is stored and ensure restricted entry of other people	94	100
Surfaces and floors are smooth, impervious, and capable of being thoroughly cleaned and disinfected	87	100
Meat is stored on hangers to avoid contamination from the floor	78.3	95
*Transportation of carcass*		
Refrigerated trucks	8.2	75
Trucks without refrigerators	27.2	25
Motorcycles (boda-boda).	84.2	85
By hand.	7.6	0
*Storage of meat*		
In refrigerators, freezers, chilled rooms	64.1	100
In a guarded glass compartment	66.3	90
On hangers	78.3	95
On the floor	2.2	0
*Staff training*		
Training for new staff	94	80
Retraining	11.4	75
HACCP-based training	7.6	75
*Government oversight*		
Monitoring/checking appropriateness of records used by your business	7.6	75
Does the government or NGO provide any form of training to your staff?	0	0
Does the government set any specific qualifications to be able to run or open a butcher shop?	2.2	20

## Data Availability

Data available on request from the authors.

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
