# Peer review of "Analysis of Food Safety Management Systems in the Beef Meat Processing and Distribution Chain in Uganda"

_foods, 2021, doi:10.3390/foods10102244_

Round 1

Reviewer 1 Report

Review ID: foods-1333395

The reviewed article describes the current (difficult) situation of the meat market in Uganda by analysing the production and distribution chain. The research included slaughterhouses, butchers and supermarkets, where health risks and safety were analysed. The title suggests a general meat market, i.e. all species, but the analysis is on beef, which should be clarified. I have no criticisms of the methodology used. The discussion of the results of the surveys is appropriate. I would propose to exclude the fragment with recommendations from Conclusions, and also try to provide prospects, including potential opportunities to improve food safety in this sector.    

Authors shall pay special attention to prepare the list of references (authors, names of journals, etc.). 

Other remarks

L53-55: Please arrange raw materials and products according to the processing: carcasses/half-carcasses/fore- or hindquarters, primal/retail cuts, raw meat products (grinding beef). Ham is produced from pigmeat.

L181: dot is missing

Table 4

‘un-edible’ replace with inedible

Description of variables without dots

Best regards

Author Response

  1. The reviewed article describes the current (difficult) situation of the meat market in Uganda by analysing the production and distribution chain. The research included slaughterhouses, butchers and supermarkets, where health risks and safety were analysed. The title suggests a general meat market, i.e. all species, but the analysis is on beef, which should be clarified.

Reply:  We thank the reviewer for the insightful comment. The title has been modified to reflect the beef market as suggested.   

  1. I have no criticisms of the methodology used. The discussion of the results of the surveys is appropriate. I would propose to exclude the fragment with recommendations from Conclusions, and also try to provide prospects, including potential opportunities to improve food safety in this sector.   

ReplyWe have enhanced the concussions as suggested.

  1. Authors shall pay special attention to prepare the list of references (authors, names of journals, etc.). 

              Reply: We have corrected the list of references as per the requirements of this journal.

  1. Other remarks
    • L53-55: Please arrange raw materials and products according to the processing: carcasses/half-carcasses/fore- or hindquarters, primal/retail cuts, raw meat products (grinding beef). Ham is produced from pigmeat.

Reply: Corrected as suggested.

  • L181: dot is missing

Reply: Corrected.

  • Table 4

‘un-edible’ replace with inedible. Description of variables without dots

Reply: Corrected as suggested.

Reviewer 2 Report

In this paper, authors reported that food safety management (FSM) practices in the Ugandan meat supply chain. The low food safety performance in the supply chain was primarily attributed to poor sanitation, hygiene, and handling practices. Therefore, Hazard Analysis and Critical 25 Control Point (HACCP)-based training and robust preventive, intervention, and monitoring systems are needed in the Ugandan meat supply chain to benefit public health and increase competitiveness. However, this study is not rigorous and the idea is not enough novel, and provide little new insight to the readers in the discipline. In addition, the section of Conclusion is digression, especially in logically arranged descriptions. In my opinion, the manuscript has not meet the publication standard of Foods. 

Author Response

In this paper, authors reported that food safety management (FSM) practices in the Ugandan meat supply chain. The low food safety performance in the supply chain was primarily attributed to poor sanitation, hygiene, and handling practices. Therefore, Hazard Analysis and Critical 25 Control Point (HACCP)-based training and robust preventive, intervention, and monitoring systems are needed in the Ugandan meat supply chain to benefit public health and increase competitiveness. However, this study is not rigorous and the idea is not enough novel, and provide little new insight to the readers in the discipline. In addition, the section of Conclusion is digression, especially in logically arranged descriptions. In my opinion, the manuscript has not meet the publication standard of Foods. 

Reply: We thank the reviewer for the comments. We have made substantial improvements in the revised manuscript which we hope would address the concerns raised by the reviewer.

Round 2

Reviewer 2 Report

The idea of this revision is not enough novel, and provide little new insight to the readers in the discipline. This revision is still not to be accepted.